# Temporal and spatial dynamics of *Plasmodium falciparum* clonal lineages in Guyana

**Mathieu Vanhove**[1,2], **Philipp Schwabl**[1,2], **Collette Clementson**[3], **Angela M. Early**[1,2], **Margaret Laws**[1,2], **Frank Anthony**[3], **Célia Florimond**[4], **Luana Mathieu**[4], **Kashana James**[3], **Cheyenne Knox**[1,2], **Narine Singh**[3], **Caroline O. Buckee**[1], **Lise Musset**[4], **Horace Cox**[3,5], **Reza Niles-Robin**[3], **Daniel E. Neafsey**[1,2]*

**1** Department of Immunology and Infectious Diseases, Harvard T.H. Chan School of Public Health, Boston, Massachusetts, United States of America, **2** Infectious Disease and Microbiome Program, Broad Institute of MIT and Harvard, Cambridge, Massachusetts, United States of America, **3** National Malaria Program, Ministry of Health, Georgetown, Guyana, **4** Laboratoire de Parasitologie, World Health Organization Collaborating Center for Surveillance of Antimalarial Drug Resistance, Center Nationale de Référence du Paludisme, Institut Pasteur de la Guyane, Cayenne, French Guiana, **5** Caribbean Public Health Agency, Port of Spain, Trinidad and Tobago

* neafsey@hsph.harvard.edu

**Data Availability Statement:** Illumina-generated short-read sequence data has been deposited in the NCBI Sequence Read Archive under BioProject PRJNA809659.

## Abstract

*Plasmodium* parasites, the causal agents of malaria, are eukaryotic organisms that obligately undergo sexual recombination within mosquitoes. In low transmission settings, parasites recombine with themselves, and the clonal lineage is propagated rather than broken up by outcrossing. We investigated whether stochastic/neutral factors drive the persistence and abundance of *Plasmodium falciparum* clonal lineages in Guyana, a country with relatively low malaria transmission, but the only setting in the Americas in which an important artemisinin resistance mutation (*pfk13* C580Y) has been observed. We performed whole genome sequencing on 1,727 *Plasmodium falciparum* samples collected from infected patients across a five-year period (2016–2021). We characterized the relatedness between each pair of monoclonal infections (n = 1,409) through estimation of identity-by-descent (IBD) and also typed each sample for known or candidate drug resistance mutations. A total of 160 multi-isolate clones (mean IBD $\geq$ 0.90) were circulating in Guyana during the study period, comprising 13 highly related clusters (mean IBD $\geq$ 0.40). In the five-year study period, we observed a decrease in frequency of a mutation associated with artemisinin partner drug (piperaquine) resistance (*pfcrt* C350R) and limited co-occurence of *pfcrt* C350R with duplications of *plasmepsin 2/3*, an epistatic interaction associated with piperaquine resistance. We additionally observed 61 nonsynonymous substitutions that increased markedly in frequency over the study period as well as a novel *pfk13* mutation (G718S). However, *P. falciparum* clonal dynamics in Guyana appear to be largely driven by stochastic factors, in contrast to other geographic regions, given that clones carrying drug resistance polymorphisms do not demonstrate enhanced persistence or higher abundance than clones carrying polymorphisms of comparable frequency that are unrelated to resistance. The use of multiple artemisinin combination therapies in Guyana may have contributed to the disappearance of the *pfk13* C580Y mutation.

**Funding:** This work was supported, in whole or in part, by the Bill & Melinda Gates Foundation [INV-009416] by DEN. Under the grant conditions of the Foundation, a Creative Commons Attribution 4.0 Generic License has already been assigned to the Author Accepted Manuscript version that might arise from this submission. This study was also supported with federal funds from the National Institute of Allergy and Infectious Diseases, National Institutes of Health, Department of Health and Human Services, under Grant Number U19AI110818 to the Broad Institute and managed by DEN. These authors received salary from the Gates grant: MV, PS, CC, AME, ML, CK, COB, HC, RN-R, DEN. These authors received salary from the U19 NIH grant: MV, PS, AMe, ML, CK, DEN. The funders had no role in study design, data collection and analysis, decision to publish, or preparation of the manuscript.

**Competing interests:** The authors have declared that no competing interests exist.

## Author summary

Malaria is caused by eukaryotic *Plasmodium* parasites, which undergo sexual recombination within mosquitoes. In settings with low transmission, such as Guyana, these parasites often recombine with themselves, leading to the propagation of identical clones. We explored the population genomics of *Plasmodium falciparum* malaria parasites in Guyana over five years to characterize clonal transmission dynamics and understand whether they were influenced by local drug resistance mutations under strong selection, including *pfk13* C580Y, which confers resistance to artemisinin, and *pfcrt* C350R, which confers resistance to piperaquine. Based on whole genome sequencing results from 1,409 monoclonal (single-strain) samples, we identified 160 multi-isolate clones, in which all parasites share at least 90% of their genomes through recent common ancestry. We observed a decrease in frequency of the *pfcrt* C350R mutation, as well as the disappearance of *pfk13* C580Y. Our findings contrast with the deterministic rise of drug resistance mutations observed in other geographic regions, sometimes associated with clonality. The simultaneous use of at least two different artemisinin combination therapies may have prevented the spread of an artemisinin-resistant clone in Guyana, suggesting a strategy for resistance management in other geographic regions.

## Introduction

Genomic data from pathogens, vectors, and/or human hosts can complement traditional epidemiological data on disease incidence and prevalence to inform decisions regarding control. In the case of malaria, several distinct use cases for genomic epidemiology have been previously identified [1], including the identification of imported cases and transmission hotspots [2,3], as well as informing strategies for local disease elimination by documenting connectivity among parasite populations mediated by human movement [4]. Most importantly, genomic data from malaria parasites can play an important role in surveillance of emerging drug resistance markers [5]. Resistance has arisen to every widely deployed antimalarial [6], and molecular surveillance has been endorsed by the WHO as a core intervention for maintaining the efficacy of current malaria drug treatment regimens [7].

Genetic surveillance of drug or insecticide resistance is typically conducted using genotyping data from specific polymorphisms associated with resistance [4,8]. However, whole genome sequencing (WGS) data and genome-wide genotyping assays can inform understanding of the context for the origin and spread of mutations, especially in cases where compensatory or epistatic mutations are required to generate a high-fitness resistance genotype capable of spreading quickly [9]. While measurable phenotypic resistance may be conferred by individual mutations, other genomic changes are often required for those mutations to be evolutionarily successful, with examples in *Plasmodium* malaria parasites [10–12], bacteria [13] and other pathogens [14].

Resistance has been arising in a small number of specific geographic locations to artemisinin (ART), which is administered with one or more partner drugs as artemisinin combination therapy (ACT) as the first line treatment for malaria caused by *Plasmodium falciparum* in most of the world. Delayed parasite clearance following ACT treatment was first observed in the Greater Mekong Subregion (GMS) of Southern Asia in early 2000s [15,16]. More recently, mutations associated with reduced susceptibility to artemisinin have also been detected in East Africa [17–21] and Papua New Guinea [22]. The most important artemisinin resistance mutation, a cysteine (C) tyrosine (Y) amino acid substitution at codon 580 (C580Y) in the propeller

domain of a kelch-domain-containing protein on chromosome 13 (*pfk13*) was first observed in the Americas in samples collected in Guyana in 2010, where five out of 94 symptomatic cases were found to carry the *pfk13* C580Y mutation [23]. In 2014, a therapeutic efficacy study (TES) from Guyana failed to detect clinical artemisinin resistance [24], but sample size was likely too low to recruit subjects with low-frequency resistance mutations. *The pfk13* C580Y mutation was observed in 14 out of 854 clinical samples in a resistance surveillance study conducted in Guyana from 2016–2017, and through whole genome sequencing we determined that all of these samples represented a single clonal parasite lineage, despite being observed in disparate regions of the country [25].

This observation of a single clonal background for the *pfk13* C580Y mutation in Guyana was unexpected because *P. falciparum* is a eukaryotic parasite that undergoes sexual recombination in mosquitoes as an obligatory component of its life cycle. However, when a mosquito bites a human host with a monoclonal infection (caused by a single parasite genomic lineage), parasites do not have an opportunity to undergo sexual outcrossing in the mosquito, and instead perform selfing, resulting in the perpetuation of the genomic lineage present in the previous human host. Malaria transmission levels are low in Guyana relative to many settings in sub-Saharan Africa, and therefore most infections are monoclonal, resulting in frequent clonal transmission. Therefore, a null hypothesis to explain the observation of *pfk13* C580Y on a single clonal background could simply invoke low transmission in Guyana as a causal mechanism.

However, a plausible alternative hypothesis is that the *pfk13* C580Y mutation was observed on a single clonal background because that genomic lineage contained important compensatory or epistatic mutations, related to the phenotype of artemisinin resistance directly or resistance to one or more partner drugs commonly administered in ACTs. Historically, resistance to antimalarials has originated *de novo* in low-transmission settings like Southeast Asia or the Americas and has only later spread to sub-Saharan Africa where malaria is much more common [26], leading to the hypothesis that low sexual outcrossing rates in such settings could facilitate the emergence of high-fitness resistance genotypes by preserving key combinations of alleles (in addition to factors such as lower immunity and higher drug pressure). Clonality has been associated with the emergence of *pfk13* C580Y in Cambodia and its subsequent spread throughout the eastern GMS [16,26], perhaps facilitated by additional mutations in this lineage (*plasmepsin 2* (*pfpm2*—PF3D7_1408000) and/or *plasmepsin 3* (*pfpm3*—PF3D7_1408100) gene amplification) that confer resistance to an important artemisinin partner drug. In East Africa, studies from Uganda [19] and Eritrea [27] reported evidence of emergence of resistance through clonal propagation with an increase in prevalence of *pfk13* mutations.

The official first-line treatment for malaria in Guyana is the ACT artemether-lumefantrine (AL), and no lumefantrine (LMF) resistance mutations are known to be segregating in Guyana *P. falciparum* populations. However, an important context for malaria transmission in Guyana is among gold-miners working in forested regions who are known to frequently self-medicate with the ACT dihydroartemisinin (DHA) -piperaquine (PPQ) -trimethoprim (TMP; DHA + PPQ + TMP) tablets [28,29]. At least two mutations that are segregating in Guyana confer resistance to piperaquine: a C350R point mutation in the chloroquine resistance transporter (*pfcrt*) gene that is endemic to the Guiana Shield region and has been increasing in frequency over the last 20 years [28,30,31], and copy number amplification of the *pfpm2* and/or *pfpm3* genes [30]. The *pfcrt* C350R mutation and *plasmepsin* 2/3 amplifications interact epistatically to yield piperaquine resistance [32], adding credibility to the hypothesis that clonal transmission may be adaptive under DHA-piperaquine pressure.

In the present study, we generated whole genome sequencing data from *P. falciparum* clinical samples collected in Guyana between 2016–2021 to profile the temporal and spatial

dynamics of clonal parasite lineages. We identify circulating clonal components (referred to as clones), defined as groups of genomically indistinguishable parasites identified under a graph-based framework [33], and we explore whether limited sexual outcrossing may have been conducive to the *de novo* origin of the *pfk13* C580Y mutation in Guyana. We specifically explore the representation of *pfk13* C580Y, *pfcrt* C350R, and *pfpm2/3* gene amplifications in clonal and unique parasite genomic backgrounds, and their co-occurence in prevalent vs. rare clonal lineages. Further, we profile new signatures of selection in the local parasite population using this deep population genomic dataset to determine whether previously uncharacterized mutations may also drive clonal dynamics, or whether the persistence and prevalence of clonal lineages in Guyana are driven by stochastic factors.

## Results

### Temporal and spatial clonal dynamics in guyana

We performed selective whole-genome amplification (sWGA) on 1,727 samples collected from Guyana between 2016 and 2021 across three time periods (Fig 1). A total of 264 genomes (15.3%) did not meet the quality criteria of at least 30% of the genome covered at $\geq$ 5-fold coverage, resulting in 1,463 samples suitable for analysis. Of this set, 54 samples were classified as multiclonal infections ($F_{ws} < 0.7$) and were excluded from subsequent analyses. The final

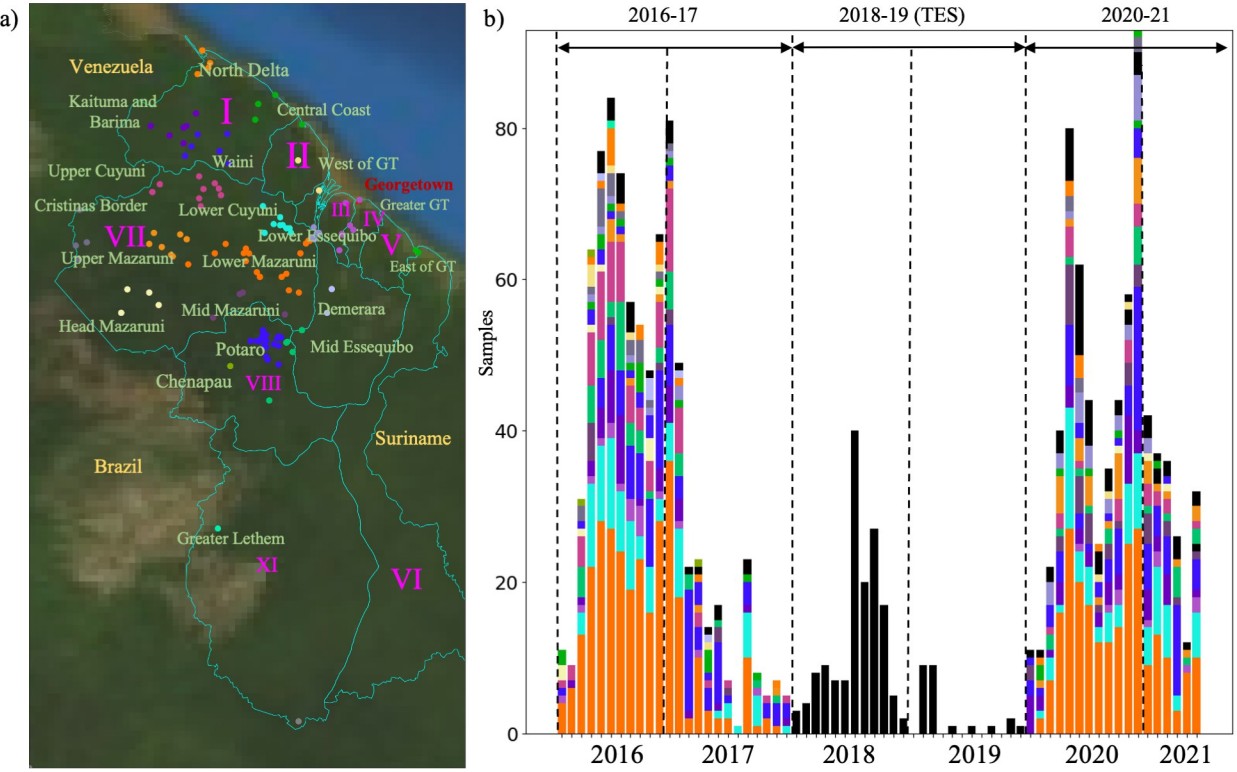

**Fig 1. Spatial and temporal distribution of *Plasmodium falciparum* samples in Guyana.** a) Epidemiological zones (n = 20) delimited using an informed approach following access using roads and rivers. Administrative Regions are indicated in purple. b) temporal distribution of samples (n = 1,409 monoclonal infections) colored by patient travel history. Three sampling periods could be observed 2016–2017 (n = 773), 2018/2019 where samples were collected as part of a therapeutic efficacy study (TES) (n = 174) with no information on patient travel history, and 2020–2021 (n = 531). Samples with no travel history information are represented in black. Map was created using Basemap v1.4.1. plotted using Guyana—Subnational Administrative Boundaries acquired from [75].

dataset for relatedness analysis contained 1,409 monoclonal genomes (S1 Table) with an average pairwise IBD across the entire dataset of 0.283 (SD = 1.114 –Fig A in S1 Text). The final dataset obtained was composed of 736 genomes from 2016–2017; 130 genomes from 2018–2019 that were collected as part of a TES; and finally 523 genomes from patient samples collected in 2019–2021. To explore patterns of relatedness due to shared recent common ancestry, pairwise identity-by-descent (IBD) values were computed between haploid genotypes (Fig 2).

Genome-wide mean IBD estimates across samples revealed patterns of shared ancestry. Network analysis identified 160 clones (C), which were defined as groups of at least two samples with a mean pairwise IBD ≥ 90%, and 332 singletons. Some of these clones formed larger highly related clusters, defined as a group of multiple clones (n ≥ 3) which displayed a mean IBD ≥ 40% and grouped together in the hierarchically-clustered dendrogram as portrayed in

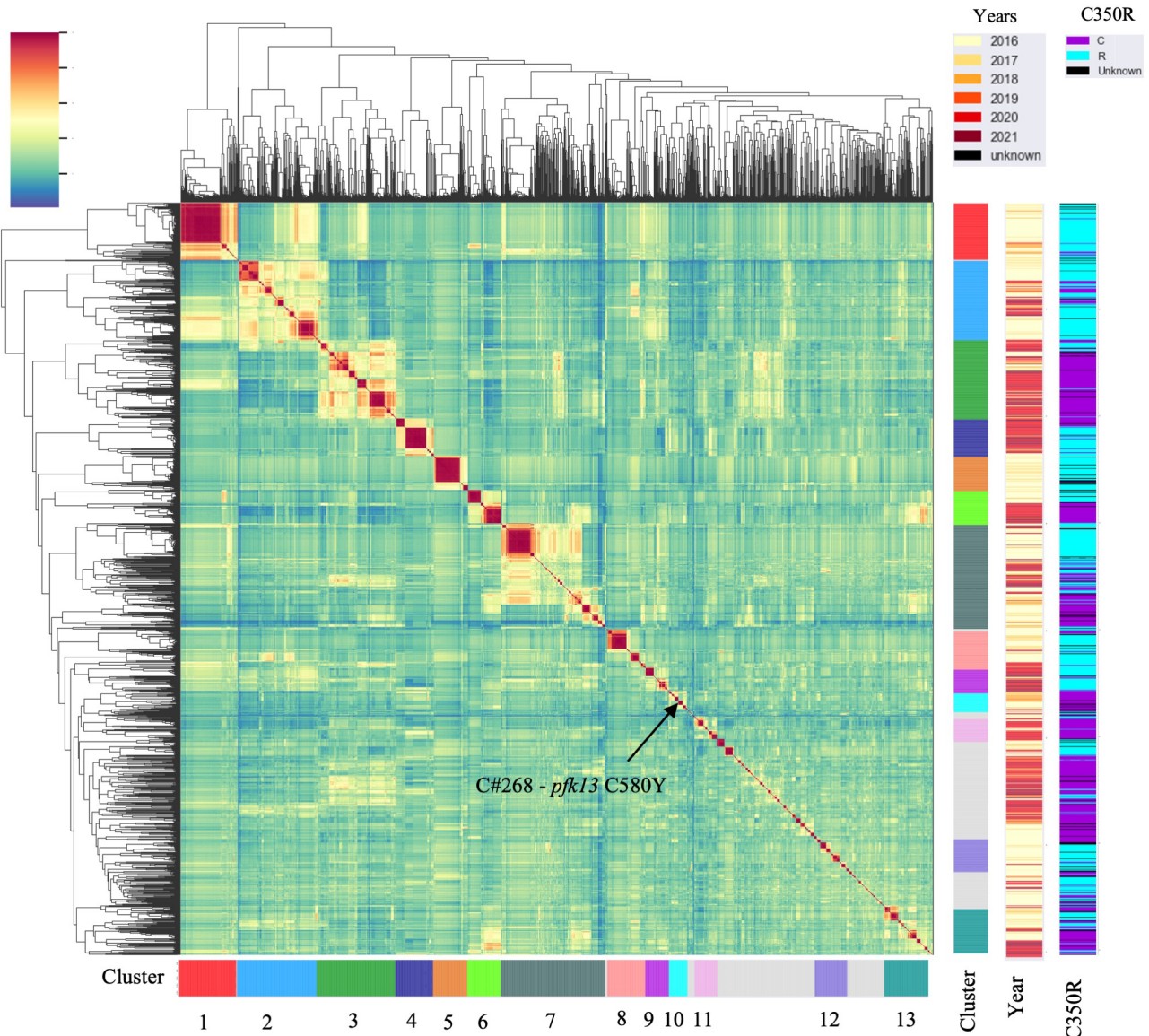

**Fig 2. The mean pairwise identity-by-descent (IBD) between samples highlighting highly related clusters (IBD ≥ 0.4).** Different sampling years are indicated as well as the presence/absence of *pfcrt* C350R.

**Table 1. Change in highly related cluster frequencies between 2016–2017 and 2020–2021.**

| Study Period | 2016–2017 | | | 2020–2021 | | |
|---|---|---|---|---|---|---|
| Highly related cluster | IBD | Number of samples | Frequency in 2016–2017 (%) | IBD | Number of samples | Frequency in 2021–2021 (%) |
| 1 | 0.867 | 88 | 11.8 | 0.957 | 2 | 0.4 |
| 2 | 0.627 | 97 | 13.0 | 0.545 | 38 | 7.2 |
| 3 | 0.708 | 25 | 3.4 | 0.534 | 107 | 20.2 |
| 4 | 0.886 | 5 | 0.7 | 0.722 | 63 | 11.9 |
| 5 | 0.763 | 64 | 8.6 | - | 0 | 0.0 |
| 6 | 0.966 | 23 | 3.1 | 0.841 | 37 | 7.0 |
| 7 | 0.476 | 124 | 16.7 | 0.409 | 49 | 9.3 |
| 8 | 0.614 | 54 | 7.3 | 0.547 | 13 | 2.5 |
| 9 | 0.578 | 3 | 0.4 | 0.566 | 39 | 7.4 |
| 10 | 0.512 | 18 | 2.4 | 0.424 | 2 | 0.4 |
| 11 | 0.485 | 8 | 1.1 | 0.541 | 32 | 6.0 |
| 12 | 0.405 | 59 | 7.9 | 0.997 | 2 | 0.4 |
| 13 | 0.525 | 51 | 6.9 | 0.421 | 37 | 7.0 |
| Other | 0.292 | 125 | 16.8 | 0.299 | 108 | 20.4 |
| Total | 0.294 | 744 | 100.0 | 0.308 | 529 | 100.0 |

Fig 2. A total of 13 highly related clusters were present in Guyana between 2016 and 2021 (Table 1). Cluster 1 was composed of seven clones and 21 singletons, including the largest clone of the study (C#1—Fig C in S1 Text), which was composed of 73 samples and disappeared in October 2018 (Fig 3). Most parasite clones in Guyana persisted for a brief time, but others lasted multiple years. The mean duration of clones was 8.3 months (251 days). Four clones belonging to different highly related clusters persisted throughout the study (C#143, C#100, C#32, C#305; Fig 3). Detection time points for the 160 clones suggested that 138 persisted for $\geq$2 months, 96 persisted for $\geq$3 months, 68 persisted for $\geq$6 months, 34 persisted for $\geq$12 months, and 7 persisted for $\geq$24 months. Seven clones were sampled over two years. A total of 69 clones were related to other clones by $\geq$ 0.40 mean IBD and highly related cluster 3 appeared as the most related to other clusters (See S1 Text for spatial analysis of clonal dynamics and Fig D in S1 Text).

### *pfk13* C580Y was restricted to a single clonal background

The C580Y mutation in the *pfk13* gene (PF3D7_1343700) was present only on single clonal background (C#268, Fig 3) as previously reported by Mathieu et al. (2020) [25]. This clone was composed of six samples and did not carry *pfcrt* C350R. The clone was part of highly related cluster 10 which was composed of six clones and five singletons (Table A in S1 Text). The clonal background harboring *pfk13* C580Y was related to clone C#270 (n = 5) and C#271 (n = 2) at mean pairwise IBD levels of 0.45 and 0.42, respectively. On average, clones circulated in 2.46 epidemiological zones and for 237.0 days. The *pfk13* C580Y-harboring cloneC#268, last observed in April 2017, was observed in six locations over 418 days (Fig 4). In terms of clonal persistence, this haplotype was among the top 20% of clones detected at multiple-time points (n = 130). We investigated nonsynonymous (NSY) mutations with a similar allelic frequency (MAF = 0.007 ± 0.05) as *pfk13* C580Y, screening 2,360 NSY mutations for their relative clonal abundance and clonal persistence (Fig 4). The temporal persistence of the *pfk13* C580Y mutation was above the mean clonal duration (t = 287.4 days, $p < 0.211$, $t_{C580Y}$ = 418.0 days) and

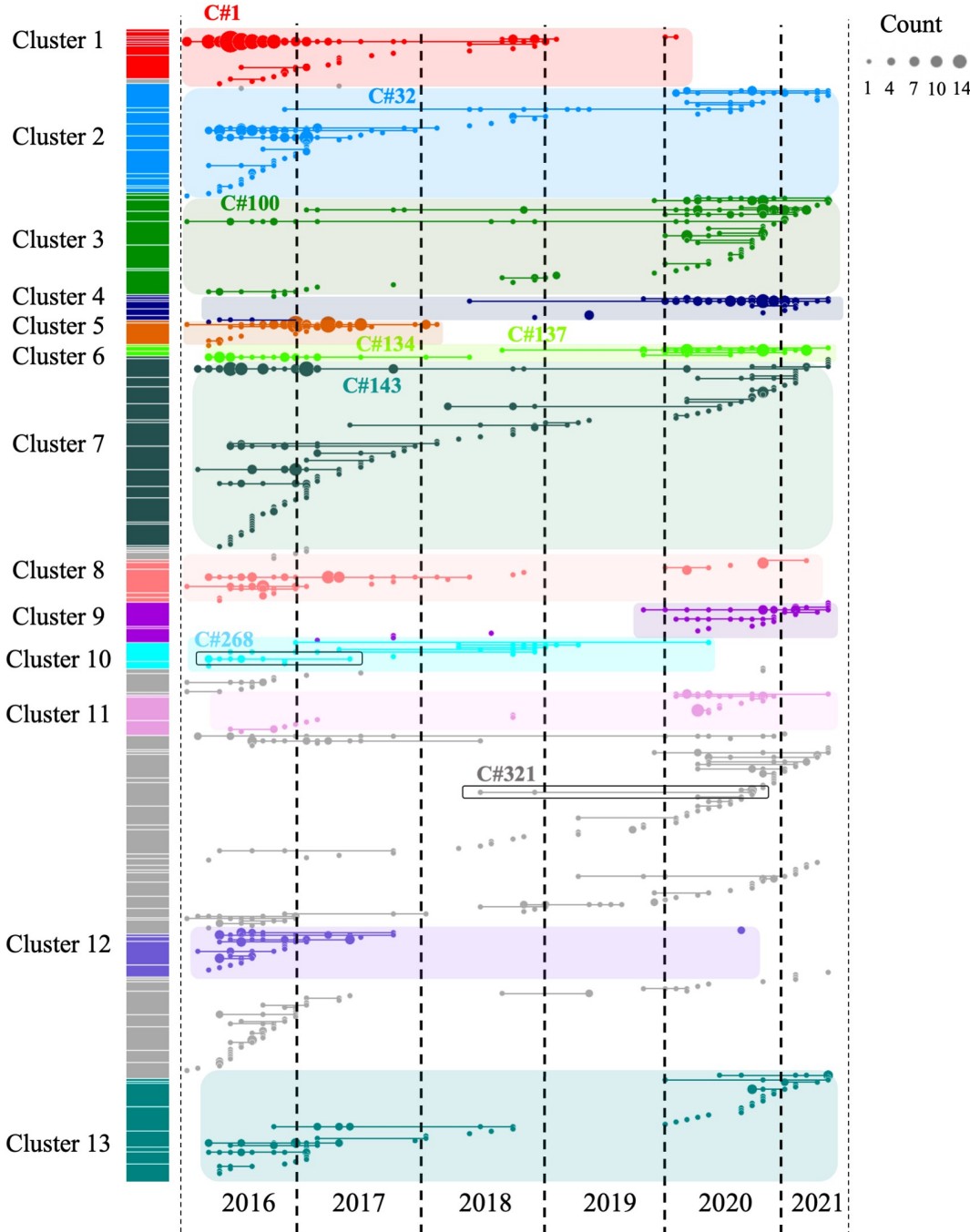

**Fig 3. Clonal temporal dynamics between 2016 and 2021.** Clone C#1 in highly related cluster 1 was the largest clone present in the dataset (n = 73 samples). C#268 is the clonal background harboring *pfk13* C580Y, while C#321 carried the *pfk13* G718S. All clones highlighted on the figure are referenced in the text.

clonal abundance was above average ($n_{C580Y}$ = 8.0, $p$ < 0.211, $n_{mean}$ = 6.0) but below the 95th percentile of polymorphisms in the same frequency class.

Two occurrences of a previously undescribed *pfk13* mutation (G718S) were also observed. The two samples were collected the same week in November 2020 in Aranka River in Region

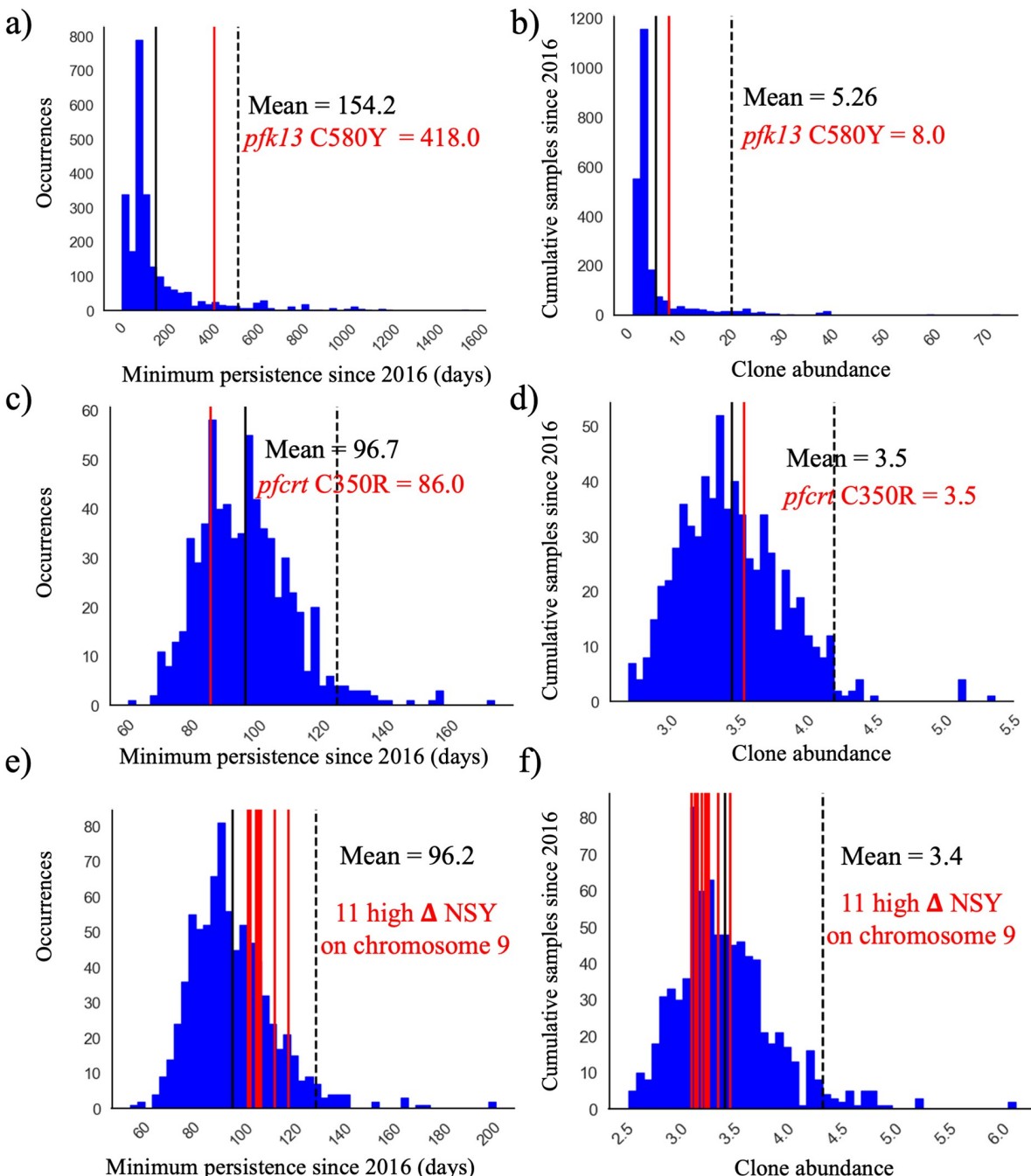

**Fig 4. Clone persistence and clonal abundance for nonsynonymous (NSY) mutation with similar (± 0.05) MAF (a-b)** *pfk13* C580Y (minor allele frequency (MAF) = 0.007 ± 0.05 n = 2,360), c-d) *pfcrt* C350R (MAF = 0.46 ± 0.05 n = 683) and (e-f) Eleven NSY (nine genes) on chromosome 9 which increased in frequency (in 99th percentile: $\Delta_{FREQUENCY} \geq 28.4\%$—MAF = 0.29 ± 0.05 n = 853). Vertical black lines represent the mean of the distribution, red vertical lines are the mutations observed, and the dashed line is 95th percentile at the particular MAF.

7. They also carried the *pfk13* K189T mutation. These samples belonged to a clonal background (C#321) composed of four samples, which was first detected in April 2018, but the other members of this clonal background did not have sufficient coverage at this position to permit allele identification.

## Decrease in *pfcrt* C350R frequency across the five year study period

In the *pfcrt* gene (PF3D7_0709000), which encodes a transmembrane digestive vacuole protein known to modulate resistance to chloroquine and other drugs [10], allelic positions 72, 76, 220, 326 (wildtype) and 356 in *pfcrt* were fixed. The frequency of *pfcrt* C350R in the dataset was 54.0% (n = 709) and was found in 222 clones (Fig 2). Additionally, six samples harbored a previously undocumented coding polymorphism in *pfcrt*: D329N. The earliest observation of the D329N mutation was obtained in September 2018 and was sampled in Georgetown as part of the TES. The D329N mutation was found in three clones (C#173, C#9, C#402), which were each composed of two samples. These samples exhibited the *pfcrt* C350 wildtype allele and were found in different highly related clusters. Between the two study periods, a change in *pfcrt* C350R frequency was observed. In 20162017, *pfcrt* C350R was present in 73.3% of samples (n = 478) while in 2020–2021, the frequency of the mutation was 36.2% (n = 191). This frequency reversion to the wildtype allele could also be observed within a highly related cluster (Table G in S1 Text). In 2016–2017, highly related cluster 6 displayed one predominant clone (C#134) carrying the *pfcrt* C350R mutation. The clone was found primarily in the Mid Essequibo zone but also appeared in six other epidemiological zones (Fig D and Table H in S1 Text. In 2020–2021, he clones composing this highly related cluster displayed the wildtype (in C#137, C#135, C#136 and C#138). These clones were still widely distributed, with C#137 occurring in nine epidemiological zones (Fig 5). Evidence was observed of multiple *de novo*

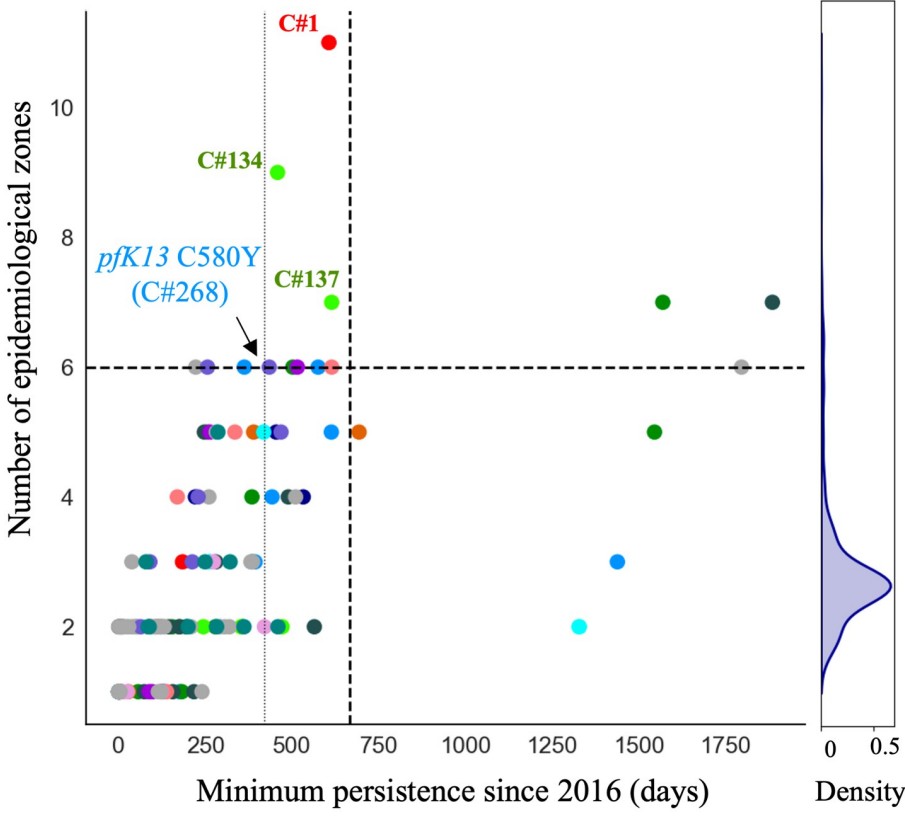

**Fig 5. Clone persistence and the number of epidemiological zones reached (> 1 sample and at least one spatial location, n = 130, mean = 287 days).** Clones are coloured by highly related clusters. C#268 carrying the *pfk13* C580Y mutation is highlighted right on the persistence 80th percentile (vertical line). The horizontal and vertical dashed lines represent the 95th percentiles. The density plot illustrates the maximum number of epidemiological zones reached by the clone. All clones labeled on the figure (C#1, C#134, C#137, C#268) are referenced in the text.

origins of the *pfcrt* C350R mutation. The wildtype and *pfcrt* C350R genotypes were observed to co-occur in nine clones. For instance, C#45 contained four samples with *pfcrt* C350R and seven representing the wildtype (Fig 6). The clone C#268 harboring the *pfk13* C580Y did not carry *pfcrt* C350R.

When investigating whether *pfcrt* C350R had an impact on clonal persistence or abundance, no significant effect was observed. The average persistence of clones carrying the mutation was 268.0 days ($p < 0.810$), while the average persistence of clones representing other nonsynonymous mutations of comparable allele frequency (MAF = 0.46 ± 0.05) was 291.0 days (Fig 4C, n = 683 NSY mutations). The abundance of clones harboring *pfcrt* C350R (mean = 6.9 samples, $p < 0.526$) was also similar to the abundance of clones representing these comparator mutations (mean abundance of 6.9 samples) (Fig 4D). Mutations significantly associated with prolonged clonal persistence included one mutation in a *falcilysin* gene (PF3D7_1360800, n = 134.5 days, $p < 0.001$) as well as two mutations in PF3D7_1133400 (*pfama1*—apical membrane antigen 1, n = 156.4 days, $p < 0.001$). Comparator NSY mutations associated with elevated average clonal abundance also included variants in *pfama1* (PF3D7_1133400, n = 4.6 samples) and *pfmsp1* (PF3D7_0930300, n = 4.3 samples).

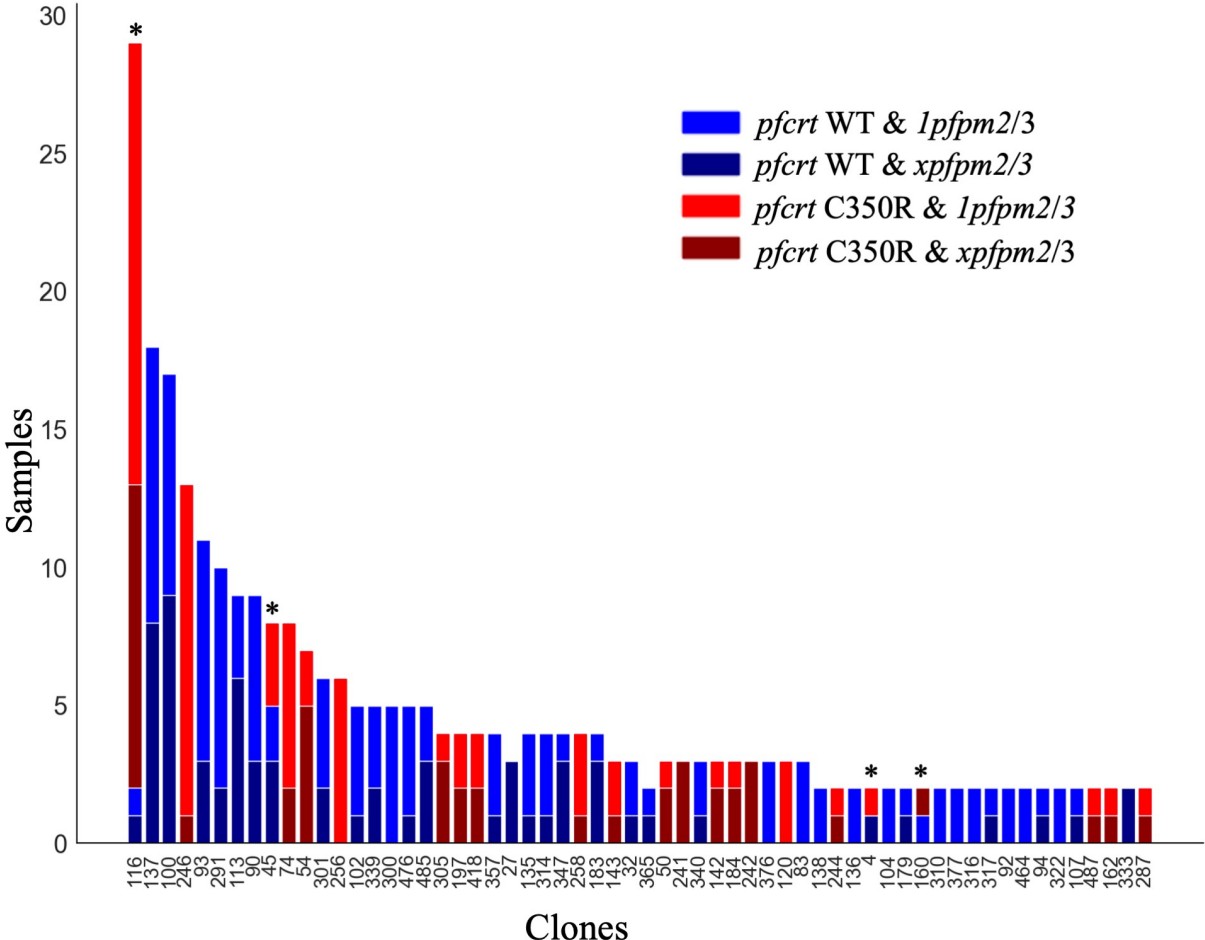

**Fig 6. *pfcrt* C350R and plasmepsin amplification (*xpfpm2/3*) in the different clones (> 1 sample) from 2020–2021.** Copy number in plasmepsin appears not consistent among clones ($p < 0.0349$) and recurrent mutational events of *pfcrt* C350R were observed in four clones (C#116, C#45, C#4, C#160). Wildtype (WT) and *pfcrt* C350R mutation are indicated in blue and red respectively. The term *xpfpm2/3* to designate the amplification of *pfpm2* or *pfpm3* and *1pfpm2/3* to denote one copy of both genes*1pfpm2/3* refer to having one copy of *pfpm2/3*.

## Co-occurrence of *plasmepsin* 2/3 duplication and *pfcrt* C350R

The combination of *pfcrt* C350R and *plasmepsin 2/3* copy number amplification has been recently demonstrated to confer piperaquine resistance in the Guiana Shield [30]. *Pfpm2/3* copy number status was estimated via qPCR from 62 samples in 2016–2017 (8.0%) [30]. Although the information was only available for a limited number of samples, a $X^2$ test revealed a significant association between *pfpm2/3* copy number and C350R ($p < 0.035$), suggesting possible selection for the epistatic resistance phenotype. We assessed *pfpm2/3* copy number for 401 samples in the 2021–2021 dataset. A total of 96 (15.2%) samples possessed *pfcrt* C350R in combination with an increase in *pfpm2/3* copy number while 87 samples (21.7%) exhibited *pfcrt* C350R with a single copy of *pfpm2/3*. Ninety-six out of 245 (39.2%) wildtype samples presented multiple copies of *pfpm2/3*. No significant association between *pfpm2/3* copy number and *pfcrt* C350R was observed during the 2021–2021 period ($X^2 = 0.89$, $p < 0.64$). Highly related clusters carried the *pfcrt* C350R mutation heterogeneously. Among clones tested for the presence of *pfpm2/3* copy number variation (CNV), the frequency of *pfpm2/3* CNV was 37.7% (60/159) (Fig 6). Only four out of the 60 clones investigated carried the duplication homogeneously, highlighting the mutational volatility of this duplication.

While the net frequency of *pfcrt* C350R decreased between 2016 and 2021, two highly related clusters exhibited a frequency increase and were predominantly carrying the mutation (Table G in S1 Text). In cluster 4, where 61 samples were observed in 2021–2021 (with only 5 samples observed across 2016–2017), 43 samples (70.5%) carried *pfcrt* C350R. Among the 33 samples with *pfcrt* C350R that were tested, 14 (42.4%) displayed *pfpm2/3* copy number amplifications, whereas 19 had a single copy. In cluster 9, which was only observed in 2021–2021, 37 samples (94.9%) carried *pfcrt* C350R (Fig E in S1 Text). In this cluster, 31 samples were tested for *pfpm2/3* copy number and only four samples (12.9%) harbored both *pfcrt* C350R and multiple *pfpm2/3* copies.

## Mutations in drug resistance genes

The rise of drug resistance polymorphisms has the potential to drive clonal dynamics. In *mdr1* (PF3D7_0523000), mutations were found at positions 1042 (n = 951; 99.8%; N1042D) and 1246 (n = 1,061;99.9%; D1246Y) with only one occurrence of the wildtype allele for each position. At position 1034, 64 samples showed S1034C (n = 844;7.6%; S1034C) and 778 samples showed double NSY mutations that restore to wildtype (n = 844–92.2%, S1034S). Two samples displayed only the non-synonymous mutation on the second codon resulting in a threonine. In *dhfr*, positions 50, 51 and 108 (PF3D7_0417200) were monomorphic while in *dhps* (PF3D7_0810800), mutations at positions 540 (n = 1,318;98.1%; K540E) and 581 (n = 1,319, 98.6%; A581G) were nearly fixed with 25 and 18 samples displaying the wildtype, respectively. A new mutation in *pfpm2*, G442H, was observed in 12 samples (1.5%) representing six clones (C#385; C#216; C#218, C#219, C#325 and C#320) and observed among different clusters.

## Shift in the selection landscape in Guyana

We searched for evidence of temporal changes in natural selection by observing the changes in allele frequencies between two time periods: 2016–2017 and 2020–2021. The frequency of highly related clusters 3, 4, 6 and 11 increased while the other clusters decreased or remained stable (Table 1 and Figs E and F in S1 Text). These changes were associated with the rise of 61 NSY mutations spread across 41 genes, which were in the 99th percentile of change in frequencies (Table B in S1 Text). These mutations included *pfk13* K189T (48.8%, 458/939 samples), which increased in frequency from 34.4% (n = 185) in 2016–2017 to 68.4% (214/313) in 2020–2021. The *pfkic6* Q1680K mutation in PF3D7_0609700 (28.9%, 319/1104) encoding a Kelch13

Interacting Candidate showed a 31.4% increase during the study period. By 2020–2021, its frequency reached 73.0% in cluster 6 and 66.7% in cluster 4 (Fig G in S1 Text). The frequency of the H3221N (30.0%,–175/583) mutation in PF3D7_1346400 (*pfvps13*) increased from 12.0% (33/275) to 48.0% (120/250) during the study period. We observed a similar increase of 34.1% for the K308E NSY mutation in PF3D7_1344000 (63.0%,636/1010) encoding an aminomethyl-transferase and a 34.4% increase of the I10F mutation (47.7%, 421/883) in the falcilysin gene (FNL) on chromosome 13 (PF3D7_1360800). In PF3D7_0701900, a *Plasmodium* exported protein, six NSY mutations were observed (Table B in S1 Text). Three NSY mutations in transcription factor *pfap2-g5* (PF3D7_1139300) showed a large increase in frequency: Q2468H increased from 28.3% (160/565) to 63.9% (205/321), G1901S from 48.8% (268/549) to 83.2% (322), and T526S from 46.5% (288/619) to 82.8% (270/326). PF3D7_0704000, encoding for a conserved *Plasmodium* membrane protein also showed a NSY mutation that increased in frequency (33.7%, 227/674 to 64.7%, 337/521).

The selection landscape of the two periods 2016–2017 and 2020–2021 was also investigated using isoRelate [34] to detect genomic regions exhibiting enhanced relatedness, with a false discovery rate of 0.01. The analysis was run on clones sampled across more than three months in 2016–2017 and in 2020–2021 ($n_{2016\text{-}2017}$ = 55 clones, $n_{2020\text{-}2021}$ = 49 clones). Selection signals (relatedness peaks) were consistent across different analysis runs with different representative samples of each clone (Fig H in S1 Text). This allowed investigation of signals of positive selection within "successful" clones to understand whether genes present on genomic segments within these clones were particularly important in Guyana. In 2016–2017, seven segments (159 genes) contained within four strong selection signals on chromosomes 2, 4, 7 and 9 were identified among long-lasting clones (see Fig 7 and see S1 Text for details). On chromosome 9 (chr9: 61,342–208,725 and 318,311–432,047), the selection signal found in long-lasting clones in 2016–2017 was observed in both short and long-lasting clones in 2020–2021. Nine genes (11 mutations) exhibited a large increase in mutation frequencies ($\Delta_{\text{FREQUENCY}} \geq 28.4\%$, Table B in S1 Text) between 2016 and 2021: PF3D7_0902400 and PF3D7_0902500 (two serine/threonine protein kinase part of the FIKK family gene), PF3D7_0903300 (unknown function), PF3D7_0904200 (PH domain-containing protein), and PF3D7_0905500 (unknown function).

## Discussion

In this study, we profiled the clonal dynamics of *P. falciparum* over a five year study period using the largest whole genome sequencing dataset yet produced for this parasite species from a single country. In contrast to the GMS and East Africa, where clonal transmission and enhanced population relatedness were directly related to the emergence of mutations conferring resistance to ACTs [27,35], Guyana offers a different perspective on clonal dynamics in the context of drug resistance emergence in a low-transmission setting. As suggested by the general lack of impact of resistance mutations on clonal persistence and abundance (Fig 3), stochastic processes creating the conditions for intermittent recombination appear to be the dominant mechanism driving clonal dynamics, rather than a selective advantage obtained from particular polymorphisms favoring a specific clonal background.

### Impact of artemisinin on clonal dynamics in Guyana

Resistant lineages can circulate at low frequencies for years before becoming dominant. In this study, a total of 160 clones aggregating into 13 highly related clusters were observed. Two highly related clusters present at the beginning of the study disappeared by 2020, while four highly related clusters increased in frequency (Table 1). Malaria transmission in the Guiana Shield is largely driven by mobile populations working in gold mining or other forest-

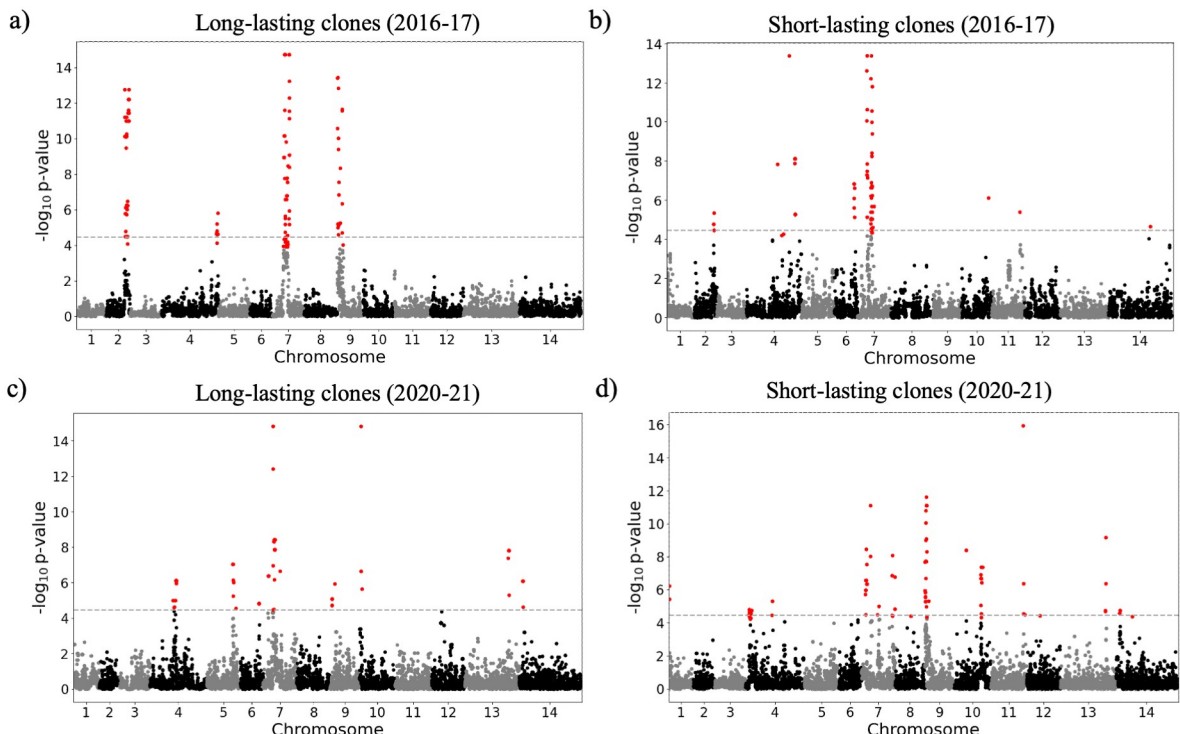

**Fig 7. Selection signals from isoRelate in long-lasting clones (sampled over three months) and short-lived clones sampled over two study periods (a-b) 2016–2017 and (c-d) 2020–2021.** Dashed lines represent the threshold for the different selection signals investigated using a false discovery rate of 0.01.

associated professions [36]. Evidence of clonal dispersal among epidemiological zones was best represented by highly related cluster 6, which has two clones spreading to seven and nine epidemiological zones in a limited amount of time (Fig 5). In 2016–2017, one clone (C#134) dominated and was primarily found in the Mid Essequibo zone. By 2021, C#134 had disappeared and a related clone (C#137) was circulating predominantly in the Potaro zone. We must note, however, that this dataset was assembled through different sampling schemes performed at different health centers and parasite origin inferred from patient travel recollection may not be consistently precise. Regions with high mining activities might also be overrepresented in travel histories (Fig E in S1 Text).

In 2004, Guyana was the first country on the continent to implement artemether-lumefantrine. Therefore, constant artemisinin pressure and shifting exposure to lumefantrine, piperaquine, and perhaps other partner drugs have imposed heterogeneous selective pressure on *P. falciparum* lineages. However, only limited evidence of allelic change potentially in response to artemisinin drug pressure has been observed. A *pfkic6* (PF3D7_0609700) Q1680K polymorphism increased in frequency by 31.65% (Table C in S1 Text) in the five year study period. The NSY mutation was present in clones that persisted longer than average ($\Delta = 31.8$ days, $p < 0.001$) and *pfkic6* is a gene which could potentially play a role in ART resistance given its association with the resistance-associated Kelch13 protein [37,38]. Other polymorphisms that appeared to be favored in Guyana were associated with potential resistance to artemisinin (Table B in S1 Text). For instance, two genes from the FIKK family (serine/threonine protein kinase: PF3D7_0902400 and PF3D7_0902500) are implicated in erythrocyte remodeling

[38,39]. Epigenetic regulation of cytoadherence linked asexual genes (*pfclag8*—PF3D7_0831600) allows parasites to develop resistance to some toxic compounds [40] Their prevalence should be closely monitored, but polymorphisms driving resistance in other geographic regions for instance nonsynonymous polymorphisms in *pffd* (ferredoxin—PF3D7_1318300), *pfcrt* or *pfarps10* (apicoplast ribosomal protein S10—PF3D7_1460900) did not show significant evidence of selection [41].

## Selection by artemisinin partner drugs

The emergence of drug resistance in the Guiana shield is of concern, considering that resistance to chloroquine and sulfadoxine-pyrimethamine emerged almost simultaneously and in an independent manner in both South America and Southeast Asia [11,42]. In Guyana, 54% of gold miners self-medicate to treat fever using DHA+PPQ+TMP tablets before seeking care [43]. The association of the *pfcrt* C350R allele with an amplification of plasmepsin (*xpfpm2/3*) has been shown to strengthen resistance phenotypes to piperaquine [28,30]. In the current study, we observed a reduction in frequency of *pfcrt* C350R from 73.0% to 24.2% across five years indicating a potential reduction in piperaquine pressure. We can speculate that a change in dominant ACT therapy from DHA+PPQ+TMP to artemether-lumefantrine could have occurred. In Cambodia, a similar decoupling of the association of *pfcrt* mutations and *xpfpm2/3* was also observed, where a shift from DHA-PPQ to artesunate-mefloquine led to a decrease in prevalence of a selected *pfcrt* mutant [44]. Frequent use of DHA+PPQ+TMP during a period of high prevalence of the *pfcrt* C350R mutation could have contributed to the emergence of the *pfk13* C580Y mutation. Subsequent increase in the use of artemether-lumefantrine may have reduced the pressure to maintain *pfcrt* C350R, and eliminated the *pfk13* C580Y mutation primarily through efficacy of lumefantrine.

Further potential evidence for reduced DHA+PPQ+TMP self-medication in recent years is the reduced prevalence of parasite genomes containing both *pfcrt* C350R and plasmepsin duplication. In Southeast Asia, an increase in copy number of the plasmepsin 2 (*pfpm2*) and/or plasmepsin 3 (*pfpm3*) genes is associated with piperaquine resistance [35,45]. These copy number amplifications have been observed to enhance piperaquine resistance *in vitro* through epistatic interaction with the *pfcrt* C350R mutation [30]. As observed in French Guiana [30], we found multiple mutational events for *pfcrt* C350R occurring within a short timespan. *Plasmepsin* duplication was also highly mutationally volatile, varying within and among conserved clonal lineages (Fig 6). The frequency of these phenomena unique to this part of the world make it difficult for a clone to thrive to the extent observed in the GMS. Gene copy number amplification may appear as a strategy for regulating expression under environmental stresses [46]. In this context, the plasticity of *pfpm2/3* might reflect a more rapid adaptation of the parasite in response to heterogeneous drug exposure. For instance, in highly related cluster 4, the dominant highly related cluster circulating in Lower Mazaruni in 2021–2021, 14 of 45 samples displayed both *pfcrt* C350R and *xpfpm2/3*, which might reflect localized recent selection by piperaquine.

## Other candidate variants associated with clonal dynamics

Although this study primarily attributes recent spatiotemporal dynamics of parasite clones in Guyana to stochastic processes in a low transmission setting rather than to selection for the preservation of specific multi-locus haplotypes, we do not suggest that meaningful selective processes are entirely absent. For instance, the clonal background containing *pfk13* C580Y was observed in six epidemiological zones across 418 days, whereas the average clone was found in 2.36 epidemiological zones and lasted on average 287 days (Fig 4). It is therefore possible that

the *pfk13* C580Y mutation improved clone fitness for a period of time. We also noted the previously unobserved *pfk13* G718S mutation in C#321, a further sign of autochthonous *pfk13* polymorphism in Guyana (Fig 3). Furthermore, we observed persistent and large clones carrying two NSY mutations in *pfama1* as well as a NSY mutation (MAF = 0.46 ± 0.05) in the falcilysin gene (PF3D7_1360800) (Tables C and D in S1 Text). The latter additionally featured among the 61 NSY mutations which increased in frequency between 2016 and 2021. Falcilysin is a metalloprotease believed to be involved in hemoglobin digestion, and has been found to be a target of chloroquine, which inhibits its proteolytic activity [47]. Given that degraded products of hemoglobin activate ART [48], it is possible that this polymorphism interferes with parasite clearance.

The outcrossing rate in Guyana appears to maintain sufficient haplotypic diversity in the population to prevent the long-term dominance of specific clones. However, four clones were sampled over four years, indicating the possibility of longer-term clonal persistence in the region. The selection signal observed at *pfcrt* was conserved throughout the dataset as previously described in global *P. falciparum* populations [34] (Fig 4). These results suggest that selection may yet be influencing clonal dynamics in Guyana, even if the impact of selection is not as stark as in the GMS [9].

Other NSY mutations that increased in frequency tended to be associated with gametocyte maturation, a process which is key to withstanding artemisinin pressure [49] because artemisinin clears only asexual parasites. Moreover, gametocyte production ultimately determines fitness because they are required for transmission. Three polymorphisms were found in transcription factor *pfap2-g5* (PF3D7_1139300). Apicomplexan-specific ApiAP2 gene family is a well-known regulator of sexual commitment and gametocyte development [50–52]. The gene appears as an important mechanism during the maturation of sexual stages through gene repression combined with other chromatin-related proteins [53]. Transcription factors (AP2 genes) involved in the gametocyte development have been previously found to display the strongest signatures of selection in French Guiana [54]. Seven other genes which increased in frequency are also related to gametocyte development. For instance, PF3D7_0904200 (PH domain—containing protein) transcripts have been shown to be enriched in gametocytes [55] and PF3D7_1474200 was found to be highly expressed in late-stage gametocytes [56].

## Relevance of *pfK13* C580Y mutation disappearance

Guyana represents the first country where a validated ART resistance mutation has appeared and then subsequently disappeared rather than increase in frequency. The *pfk13* C580Y mutation was restricted to a single clonal background and was last observed in April 2017. This clonal background lacked the *pfcrt* C350R mutation, making it likely susceptible to PPQ, which we may infer has been subject to decreasing use (from declining resistance allele frequencies) through self-medication and might have led to this disappearance in the presence of efficacious artemether-lumefantrine treatment. Previous TES in the region have hinted at resistance to artemether-lumefantrine [57] and artesunate monotherapy [56,58], but evidence from TES in Guyana is lacking. A modeling study exploring factors associated with the spread of *pfk13* mutations found that deploying multiple first-line therapies was an effective approach to postponing treatment failure [59]. The simultaneous use and potentially shifting balance of at least two ACTs in Guyana might have therefore led to the elimination of the *pfk13* C580Y mutation and its clonal background.

Clonal turnover in Guyana appears to be different from the patterns observed in other regions like South-East Asia and East Africa. In the eastern GMS, artemisinin was initially used as monotherapy facilitating rapid resistance expansion via hard selective sweep [60].

These observations indicate that drug resistance emergence does not result in the same patterns of clonal dynamics in different geographic locations, perhaps due to unique differences in disease epidemiology and drug pressure across settings. Further molecular surveillance of clonal dynamics is warranted in settings where it occurs, given the potential association of clonal transmission with both known and novel mutations associated with drug resistance.

## Materials and methods

### Ethics statement

Written informed consent was obtained from all study participants or their parents or guardians. The analysis of the samples was also approved by the Environmental Protection Agency in the frame of the Nagoya Protocol on Access to Genetic Resources and the Fair and Equitable Sharing of Benefits Arising from their Utilization. Regarding 2018–2021 samples, this study was approved by a local ethical committee in Guyana (Institutional Review Board, no. 645/2019) as well as the Harvard Longwood IRB (IRB19-1779).

### Sample collection and mapping of epidemiological zone

We evaluated 1,727 clinical samples collected from symptomatic individuals seeking treatment and diagnosed with malaria infection between 2016 and 2021 who provided informed consent for genetic analysis of their parasite samples. Samples were collected in accordance with ethical requirements (S1 Text) and Participants provided informed consent in accordance with the ethical regulations of the countries. Collection followed informed consent using a study protocol approved by ethical committees of Harvard University and the government of Guyana. Samples were collected as dried blood spots on Whatman FTA cards. Samples dating from 2016–2017 (n = 837) were collected for a resistance surveillance project [25]. Samples dating from 2018–2019 (n = 174) were collected in the context of a therapeutic efficacy study (TES). Samples dating from 2020–2021 (n = 716) were collected for a separate malaria molecular surveillance study from individuals diagnosed with *P. falciparum* infection (Fig 1). Blood spot collection was performed for microscopy or RDT-positive malaria cases primarily in Georgetown, Bartica, Port Kaituma, Mahdia, and Lethem at medical facilities. Infection localities were inferred based on voluntary responses to the query: "Where [the] patient stayed 2 weeks ago". Each travel history response was classified to one of 460 country-wide localities with known latitude/longitude coordinates. Participants provided informed consent in accordance with the ethical regulations of the countries and written consent was obtained by the Participants.

To define epidemiological zones, we first matched travel history responses to a catalog of malaria survey sites used by the Guyana Ministry of Health (MoH). We then mapped survey sites onto a custom shape file summarizing the country's primary river and road coordinates and onto a raster map of motorized transport resistance [61] available at https://malariaatlas.org/ as in [62]. Sites were clustered based on river/road connectivity in the R package 'riverdist' [63], travel conductance using the R package 'gdistance' [64], and manual assessment of coordinates on river/road and resistance layers in QGIS.

Samples were collected in specific recruitment locations and patient travel history was documented. To investigate spatial patterns, 20 epidemiological zones were defined following roads and rivers access (Fig 1A). Patient travel history revealed that a majority of infections were acquired in Lower Mazaruni River in Region 7 (n = 434, 36.1%), followed by Potaro River in Region 8 (n = 162, 13.5%), as well as along the Cuyuni River (Tables A and C in S1 Text for details on highly related clusters dispersal). Travel history data from the TES (n = 174)

conducted in 2018 and 2019 in Georgetown and Port Kaituma were not recorded. Overall, location data were missing for 216 samples (14.9%).

## Genomic data generation

DNA extraction was performed using two approaches according to year of collection. Samples from 2016-2017-2018-2019 (n = 1,011) were extracted from dried blood spots using the QIAamp DNA mini kit according to the manufacturer's instructions (Qiagen, Hilden, Germany). For samples from 2020–2021 (n = 716), we performed DNA extraction on all patient samples using a ThermoFisher blood and tissue kit and a ThermoFisher Kingfisher instrument. We performed selective whole genome amplification (sWGA) [65] on all samples to enrich the proportion of parasite DNA relative to host DNA. We performed library construction using a NEBNext kit on the enriched DNA samples and sequenced them on an Illumina NovaSeq instrument using 150 bp paired-end reads. To investigate potential batch effects between the isolates from 2016–2019 (extracted with QIAamp DNA mini kit and sequenced on Illumina HiSeqX instrument) and the ones from 2020–2021 (extracted with ThermoFisher blood and tissue kit and sequenced using Illumina NovaSeq), we quantitatively inspected coverage variations and no signs of technical artifacts were observed (Fig B in S1 Text). We aligned reads to the *P. falciparum* 3D7 v.3 reference genome assembly and called variants following the Pf3K consortium best practices (https://www.malariagen.net/projects/pf3k). We used BWA-MEM [66] to align raw reads and remove duplicate reads with Picard tools [67]. We called SNPs using GATK v3.5 HaplotypeCaller [68]. We performed base quality score and variant quality score recalibration using a set of Mendelian-validated SNPs, and restricted downstream population genomic analyses to SNPs observed in 'accessible' genomic regions determined to be amenable to high quality read alignment and variant calling [69]. Individual calls supported by fewer than five reads were removed and any variant within 5 nucleotides of a GATK-identified indel was also excluded. Samples exhibiting quality monoclonal genome data ($\geq$ 5x coverage for >30% of the genome) were included in relatedness analyses. The final dataset to investigate mutation comprised 74,357 SNPs.

## Relatedness analysis using identity-by-descent

We performed analyses of relatedness by estimating pairwise identity-by-descent (IBD) between all monoclonal patient samples (n = 1,409). We estimated IBD using the hmmIBD algorithm [70], incorporating all SNPs that were called in $\geq$ 90% of samples and with minor allele frequency $\geq$ 1%, resulting in a final set of 16,806 SNPs [70]. We used the $F_{ws}$ metric ($< 0.70$) to identify and exclude samples containing multiclonal infections [71]. We conducted subsequent analyses in Python v3.8. We constructed clones using Networkx v.2.8 [72]. clones, defined as groups of statistically indistinguishable parasites identified under a graph theoretic framework [33], were obtained using a mean IBD threshold $\geq$ 0.90. Highly related clusters were defined as a group of clones (n $\geq$ 3) which clustered together in the hierarchically-clustered dendrogram (UPGMA algorithm) performed using seaborn v0.13.0 with a threshold of 3 [73] and which also displayed a mean IBD $\geq$ 0.40. This threshold was chosen based on this specific dataset and because it represents genomes separated by 1–2 recombination events. To identify temporal changes across the sampling period, we investigated NSY SNPs that were in the 99[th] percentile of change in frequencies. To investigate whether mutations in *pfcrt* were significantly associated with longer duration or frequency of clones, we selected mutations within ± 0.05 of the minor allele frequency (MAF) of *pfcrt* C350R (MAF = 0.46). We evaluated SNP enrichment in clones with similar duration/frequency as the C350R mutation in pfcrt. Mutations within the 95[th] percentile were considered as significant.

We investigated signals of selection using the genome-wide test statistics ($X_{iR,s}$) in isoRelate v.0.1.0 [34] in R. $X_{iR,s}$ is a chi-squared distribution test statistic for measuring IBD. Briefly, an IBD matrix status with SNPs as rows and sample pairs as columns is created. A normalization procedure is implemented by subtracting the column mean from all rows to account for the amount of relatedness between each pair. Secondly, to adjust for differences in SNP allele frequencies, the row mean is subtracted from each row and divided by $p_i(1-p_i)$, where pi is the population allele frequency of SNP i. Then, row sums are computed and divided by the square root of the number of pairs. Summary statistics are normalized genome wide. To do this, all SNPs are binned in 100 equally sized bins partitioned on allele frequencies. Finally, the mean was subtracted and divided by the standard deviation of all values within each bin. Z-scores were squared to allow only positive values and such that the statistics followed a chi- squared distribution with 1 degree of freedom. We calculated $X_{iR,s}$ and obtained $-\log_{10}$ transformed p-values, and used a false discovery rate threshold of 0.05 to assess evidence of positive selection.

## Plasmepsin 2/3 copy number estimation

DNA from selected samples was used for amplification by quantitative PCR (qPCR) to estimate the copy number of plasmepsin 2 and plasmepsin 3 (*pfpm 2/3*) using a previously published protocol that does not distinguish between the two genes [45]. *P. falciparum* tubulin primers (*Pftub*) were used as a single copy comparator locus (forward-5'-TGATGTGCG-CAAGTGATCC-3'; reverse-5'-TCCTTTGTGGACATTCTTCCTC-3') and amplified separately from *pfpm* (forward-5'-TGGTGATGCAGAAAGTTGGAG-3'; reverse-5'-TGGGACCCATAAATTAGCAGA-3'). qPCR reactions were carried out in triplicate in 20 μL volumes using 384-well plates (Fisher Scientific, Hampton, NH) using 10 μL SensiFAST SYBR No-ROX mix (2x) (Bioline Inc., Taunton, MA), 300 nM forward and reverse primer, 6.8 μl nuclease-free H2O, and 2 uL DNA template as previously described by [74]. The reactions were performed using the following conditions: initial denaturation at 95 ˚C for 15 minutes followed by 40 cycles at 95 ˚C for 15 seconds, 58 ˚C for 20 seconds, and 72 ˚C for 20 seconds; a melt curve starting at 95 ˚C for 2 minutes, 68 ˚C for 2 minutes, followed by increments of 0.2 ˚C from 68 ˚C to 85 ˚C for 0:05 seconds and a final step at 35 ˚C for 2 minutes. Copy number value was calculated using the $2^{-\Delta\Delta Ct}$ method [74]. Means of *pfpm2* and *Pftub* were calculated for 3D7 (a single copy control) using six replicates. Standard deviation should not be more than 25% including all triplicates for the DNA samples. If the value was between 0.6 and 1.5, the copy number is estimated as 1, whereas if the value was between 1.5 and 2.4, the copy number estimated was 2. We use the term *xpfpm2/3* to designate the amplification of *pfpm2* or *pfpm3* and *1pfpm2/3* to denote one copy of both genes similarly to [30].

## Supporting information

**S1 Table. Sample details and genome coverage (n = 1,409).**
(XLSX)

**S1 Text.** Table A. Emerging highly related clusters emerging in Guyana. Highly related clusters are defined as a group of at least 3 clones with an average IBD ≥ 0.40. Table B. Polymorphism which increased in frequency between 2016–2017 and 2020–2021. Table C. Significant NSY mutations associated with clone persistence (within MAF = 0.46). Table D. Significant NSY mutations associated with clonal abundance (within MAF = 0.46). Table E. Genes identify in isoRelate for 2016–2017 (n = 24). Table F. Genes identify in isoRelate for 2020–2021 (n = 16). Table G. Prevalence of C350R among the different genomic clusters. Table H. Epidemiological zones associated with sample provenance. Fig A. IBD distribution in the dataset

(1,445 genomes). Fig B. Average coverage using an overlapping sliding window (size = 1000, overlap = 100) for the samples sequenced in two batches: the isolates obtained over the 2016–2019 period and the 2020–2021 period. Fig C. The mean IBD between samples highlighting highly related cluster 1—(n = 107 isolates). Fig D. Country-wide distribution of highly related clusters. Fig E. Relatedness for the two study periods– 2016–2017 and 2020–2021. Fig F. Change mutation frequencies across 2016–2017 and 2020–2021. Fig G. Haplotypes in a 10kb window for four different genes. Rows represent individual genome and columns are alleles: mutants are colored in green while wildtype is in white, grey represents missing data. Fig H. Selection signals from isoRelate in clones and singleton isolate sampled over two study periods (a-b) 2016–2017 and (c-d) 2020–2021. Fig I. Selection signals from isoRelate between clones carrying *pfcrt* C350R and wildtype clones. Fig J. Mean pairwise IBD within 100-kb overlapping windows with 10-kb overlap different chromosomes for long-lasting and short lasting clones over the two time periods. a-d) isoRelate results for the different datasets, e-f) mean pairwise IBD on chromosome 7, i-l) mean pairwise IBD on chromosome 9. Fig K. Mean pairwise IBD within 100-kb overlapping windows with 10-kb overlap different chromosomes for long-lasting and short lasting clones over the two time periods. a-d) isoRelate results for the different datasets, e-f) mean pairwise IBD on chromosome 2, i-l) mean pairwise IBD on chromosome 4. (DOCX)

## Acknowledgments

We thank the participants who contributed blood samples to this study, as well as the technicians who collected and processed the samples.

## Author Contributions

**Conceptualization:** Mathieu Vanhove, Philipp Schwabl, Angela M. Early, Daniel E. Neafsey.

**Data curation:** Mathieu Vanhove, Philipp Schwabl, Angela M. Early, Margaret Laws, Célia Florimond, Cheyenne Knox, Narine Singh, Caroline O. Buckee, Horace Cox.

**Formal analysis:** Mathieu Vanhove, Philipp Schwabl, Luana Mathieu, Cheyenne Knox, Lise Musset.

**Funding acquisition:** Collette Clementson, Margaret Laws, Frank Anthony, Kashana James, Narine Singh, Caroline O. Buckee, Lise Musset, Reza Niles-Robin, Daniel E. Neafsey.

**Investigation:** Cheyenne Knox.

**Methodology:** Mathieu Vanhove, Collette Clementson, Angela M. Early, Célia Florimond.

**Project administration:** Margaret Laws, Caroline O. Buckee, Lise Musset, Reza Niles-Robin, Daniel E. Neafsey.

**Resources:** Margaret Laws, Luana Mathieu, Caroline O. Buckee, Reza Niles-Robin.

**Supervision:** Daniel E. Neafsey.

**Validation:** Lise Musset, Daniel E. Neafsey.

**Visualization:** Mathieu Vanhove.

**Writing – original draft:** Mathieu Vanhove.

**Writing – review & editing:** Philipp Schwabl, Angela M. Early, Lise Musset, Reza Niles-Robin, Daniel E. Neafsey.

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
