## [Decision Letter · Decision Letter 0]

17 Mar 2024

Dear Dr. Vanhove,

Thank you very much for submitting your manuscript "Temporal and spatial dynamics of Plasmodium falciparum clonal lineages in Guyana" for consideration at PLOS Pathogens. As with all papers reviewed by the journal, your manuscript was reviewed by members of the editorial board and by several independent reviewers. The reviewers appreciated the attention to an important topic. Based on the reviews, we are likely to accept this manuscript for publication, providing that you modify the manuscript according to the review recommendations.

Reviewer 1 does ask for a major revision and spells out changes that would strengthen the manuscript. Reviewer 2 requests a minor revision. Upon receipt of your revised manuscript we are likely to return it to Reviewer 1 to ensure that the manuscript was fully responsive. 

Sincerely,

David A. Fidock, Ph.D.

Guest Editor

PLOS Pathogens

Margaret Phillips

Section Editor

PLOS Pathogens

Michael Malim

Editor-in-Chief

PLOS Pathogens

orcid.org/0000-0002-7699-2064

Thank you for submitting this interesting manuscript. This has now been evaluated by two expert reviewers and their comments are favorable. Reviewer 1 does ask for a major revision and spells out changes that would strengthen the manuscript. Reviewer 2 requests a minor revision. Upon receipt of your revised manuscript we are likely to return it to Reviewer 1 to ensure that the manuscript was fully responsive. Given the requests, it should be feasible to resubmit a suitably revised manuscript within 30 days. Please let us know if additional time would be required. Please contact us also if the Reviewer 1 comments, which were provided as an attached file, are not conveyed.

We look forward to receiving a revised manuscript.

Reviewer Comments (if any, and for reference):

Reviewer's Responses to Questions

**Part I - Summary**

Reviewer #1: (No Response)

Reviewer #2: In this interesting paper, the authors present the results of an analysis of >1400 P. falciparum genomes sampled from Guyana from 2016-2021 to understand the temporal and spatial dynamics of parasite clones and related clusters and their relationship to the prevalence and dynamics of drug resistance mutations in the study area. Overall, the paper provides important insights into the evolution of drug resistance in Guyana, as well as selection signals that might contribute to a favorable background for emergence of resistance to artemisinin derivatives and partner drugs. At times the results were challenging to follow and were not always directly linked to the conclusions, and the discussion and interpretation of the results could be expanded in some places. Specific suggestions/questions are noted below.

Comments:

1) The notation used in the manuscript is sometimes confusing, with clones, clonal components (same as clones?), highly related clusters, and spatial clusters. Perhaps consider calling the spatial clusters something else, so as not to confuse with the clusters based on relatedness? Phrases like clone size were also a bit confusing, since a clone itself would not have a size. I assume clone size refers to abundance of that clone?

2) The spatial analysis and results seem to be underemphasized in the manuscript in general. In addition, some clarification is needed regarding how the spatial analysis was used to infer where an individual acquired their infection. If an individual was diagnosed in one area but indicated travel to another area, how was the source of infection determined? It is very difficult to know the exact location/time of when an individual may have been infected.

3) It would be helpful for the authors to address whether there might be any “batch” effects that could confound temporal associations, given that the samples from the different time points were processed differently (i.e., samples from one time point underwent sWGA while the others did not). Some information showing similar levels of coverage, particularly in the regions of the assessed drug resistance loci would be helpful in ensuring that the temporal trends aren’t a result of technical artefacts.

4) In general, the discussion section/interpretation of the results could be expanded and/or better linked to the observed results. For example, the link between the statement in lines 301-303 that “Stochastic processes with intermittent recombination appear to be the dominant mechanism driving clonal diversity rather than a selective advantage obtained from particular polymorphisms favoring a specific clonal background” and the presented data is not completely clear. Although this may be true, it would be helpful for the authors to state explicitly which results/lines of evidence suggest that stochastic processes rather than selective advantage are driving the observed patterns.

5) Likewise, the discussion in lines 327-330, particularly the statement “Other polymorphisms that appeared to be favored in the Guyana landscape were associated with potential resistance to artemisinin…” should be expanded, as there are several mutated genes (beyond KIC6) in Table 2 that are either the same as or have overlapping function with genes identified as contributors to artemisinin resistance in the GMS. Examples include the FIKKs, CLAGs, and others.

6) Also, in the discussion of selection by ACT partner drugs, you might consider contrasting these results with the PfCRT/pfpm2 copy number dynamics observed in the GMS, where copy number decreased rapidly after reduction in DHA-PPQ pressure, but PfCRT mutation prevalence remained high (see Shrestha et al., JID, 2021). This study also showed a “decoupling” of the plasmepsin amplification and the PfCRT mutations associated with piperaquine resistance, consistent with what is observed in this study.

7) The statement in lines 416-417 is not quite accurate – artemisinin resistance in the GMS started as a soft sweep with multiple K13 mutations emerging on different genetic backgrounds – later C580Y appeared to outcompete the other variants in the eastern GMS, but not the western GMS, where different K13 mutations predominate.

Minor Points

1) Line 128 – the subheading denotes Temporal and Spatial dynamics, but the spatial dynamics are not discussed at all in this section.

2) Were all sequenced isolates from clinical infections (as opposed to asymptomatic infections observed through active surveillance)? If so, it would be good to note this in the methods section.

**Part II – Major Issues: Key Experiments Required for Acceptance**

Reviewer #1: (No Response)

Reviewer #2: (No Response)

**Part III – Minor Issues: Editorial and Data Presentation Modifications**

Reviewer #1: (No Response)

Reviewer #2: There are several typos in the supplementary material – may want to do a thorough proofread.

PLOS authors have the option to publish the peer review history of their article (what does this mean?). If published, this will include your full peer review and any attached files.

Reviewer #1: No

Reviewer #2: No

Figure Files:

Data Requirements:

Reproducibility:

References:

---

## [Decision Letter · Decision Letter 1]

24 May 2024

Dear Dr. Vanhove,

We are pleased to inform you that your manuscript 'Temporal and spatial dynamics of Plasmodium falciparum clonal lineages in Guyana' has been provisionally accepted for publication in PLOS Pathogens.

Best regards,

David A. Fidock, Ph.D.

Guest Editor

PLOS Pathogens

Margaret Phillips

Section Editor

PLOS Pathogens

Michael Malim

Editor-in-Chief

PLOS Pathogens

orcid.org/0000-0002-7699-2064

Dear Drs. Vanhove and Neafsey,

Thank you for submitting a revised version of your article on P. falciparum clonal lineages in Guyana. The resubmission has now been reviewed by reviewer 1, who noted that all comments have been addressed and that the article is substantially improved. I am happy to let you know that you article is being accepted for publication. One request, however, is that you please carefully proofread the manuscript and supplementary materials, as there are some areas with typographical errors or sentences that would improve from some grammatical revision. You can add these changes at the time of aligning the formatting as requested by the journal staff.  

Congratulations on this interesting body of work and thank you for publishing this study in PLoS Pathogens.

Reviewer Comments (if any, and for reference):

Reviewer's Responses to Questions

**Part I - Summary**

Reviewer #1: The manuscript has been greatly improved. All my comments have been addressed.

**Part II – Major Issues: Key Experiments Required for Acceptance**

Reviewer #1: (No Response)

**Part III – Minor Issues: Editorial and Data Presentation Modifications**

Reviewer #1: (No Response)

PLOS authors have the option to publish the peer review history of their article (what does this mean?). If published, this will include your full peer review and any attached files.

Reviewer #1: **Yes: **Liwang Cui

---

## [Editor Report · Acceptance letter]

8 Jun 2024

Dear Dr. Vanhove,

We are delighted to inform you that your manuscript, "Temporal and spatial dynamics of <i>Plasmodium falciparum<i> clonal lineages in Guyana," has been formally accepted for publication in PLOS Pathogens.

Best regards,

Michael Malim

Editor-in-Chief

PLOS Pathogens

orcid.org/0000-0002-7699-2064